# Modeling the effects of tree species and incubation temperature on soil's extracellular enzyme activity in 78-year-old tree plantations

Xiaoqi Zhou[1,2#], Shen S. J. Wang[3#], Chengrong Chen[2]

[1]Tiantong National Station for Forest Ecosystem Research, Center for Global Change and Ecological Forecasting, Shanghai Key Lab for Urban Ecological Processes and Eco-restoration, School of Ecological and Environmental Sciences, East China Normal University, Shanghai 200241, China

[2]Australian Rivers Institute and Griffith School of Environment, Griffith University, Nathan, Brisbane 4111, Queensland, Australia

[3]Machine Learning Systems, Computing and Information Systems, The University of Melbourne, Melbourne 3010, Australia
#These authors contributed equally to this work.

*Correspondence to*: Xiaoqi Zhou (xqzhou@des.ecnu.edu.cn); Chengrong Chen (c.chen@griffith.edu.au)

**Abstract.** Forest plantations have been widely used as an effective measure for increasing soil carbon (C) and nitrogen (N) stocks and soil enzyme activities play a key role in soil C and N losses during decomposition of soil organic matter. However, few studies have been carried out to elucidate the mechanisms behind the differences in soil C and N cycling by different tree species in response to climate warming. Here, we measured the responses of soil's extracellular enzyme activity (EEA) to a gradient of temperatures using incubation methods in 78-year-old forest plantations with different tree species. Based on a soil enzyme kinetics model, we established a new statistical model to investigate the effects of temperature and tree species on soil EEA. In addition, we established a tree species–enzyme–C/N model to investigate how temperature and tree species influence soil C/N contents over time without considering plant C inputs. These extracellular enzymes included C acquisition enzymes (β-glucosidase, BG), N acquisition enzymes (*N*-acetylglucosaminidase, NAG; leucine aminopeptidase, LAP) and phosphorus acquisition enzymes (acid phosphatases). The results showed that incubation temperature and tree species significantly influenced all soil EEA and *Eucalyptus* had 1.01-2.86 times higher soil EEA than coniferous tree species. Modeling showed that *Eucalyptus* had larger soil C losses but had 0.99-2.38 times longer soil C residence time than the coniferous tree species over time. The differences in the residual soil C and N contents between *Eucalyptus* and coniferous tree species, as well as between slash pine (*Pinus elliottii* Engelm. var. *elliottii*) and hoop pine (*Araucaria cunninghamii* Ait), increase with time. On the other hand, the modeling results help explain why exotic slash pine can grow faster, as it has 1.22-1.38 times longer residual soil N residence time for LAP, which mediate soil N cycling in the long term, than native coniferous tree species like hoop pine and kauri pine (*Agathis robusta* C. Moore). Our results will be helpful for understanding the mechanisms of soil C and N cycling by different tree species, which will have implications for forest management.

# 1 Introduction

Global mean temperature is predicted to increase by 1.8–4.0 $^{\circ}$C by the end of this century as a result of anthropogenic activities that increase carbon dioxide ($CO_2$) in the atmosphere (IPCC, 2013). Soil stocks large amounts of carbon (C) in terrestrial ecosystems, at least four times greater than that of the global stocks of C in the atmosphere and living plants (Jobbágy and Jackson, 2000). In the context of climate warming, minor losses of C via decomposition of soil organic matter (SOM) can cause positive feedback to atmospheric $CO_2$ concentrations and global temperature (IPCC, 2013), resulting in increase in plant growth and decomposition of SOM (Davidson and Janssens, 2006; Wu et al., 2011), which can profoundly alter soil C and nitrogen (N) cycling (Luo, 2007).

Establishing forest plantations has been accepted as an effective measure for increasing soil C stocks and mitigating atmospheric $CO_2$ in national budgets (Vesterdal et al., 2013). Afforestation with different tree species has been found to enhance soil C stocks (Berthrong et al., 2009), with large differences in soil C sequestration under different tree species (Vesterdal et al., 2013). Until now, however, the underlying mechanisms behind the differences in soil C and N contents under different tree species have remained unclear (Hobbie, 2015). Alongside C and N inputs via litter decomposition and root exudation by different tree species, C and N losses via SOM decomposition are important for soil C and N cycling. During the decomposition of SOM, soil's extracellular enzyme activities (EEA) represent the rate limiting step of decomposition, marking the conversion of SOM into dissolved organic C and N, which is then metabolized by microbial decomposers (Schimel and Bennett, 2004; Caldwell, 2005; Bengtson and Bengtsson, 2007; Conant et al., 2011). Given the importance of soil EEA, a soil C/N model that incorporates soil EEA is a useful measure for investigating the effect of tree species on soil C and N cycling, which may improve our understanding of the mechanisms underlying differences in soil C and N contents under different tree species.

Previous work has shown that in a variety of ecosystems, the enzymatic activities associated with decomposition differ, depending upon the quality of SOM such as soil C:N ratios (Sinsabaugh et al., 2002). Forest plantations with different tree species have been reported to have large differences in the quantity and quality of SOM, thus greatly influencing soil EEA (Lovett et al., 2004; Lu et al., 2012; Hobbie, 2015). Generally, as soil EEA increases with increasing temperature (Koch et al., 2007), to get a certain amount of substrate via decomposition of SOM, microbes in warmer soils need to produce fewer extracellular enzymes that are involved in C and nutrient cycling (Allison, 2005). However, there are only a handful of studies that have measured the responses of soil's EEA to warming under different tree species (Allison et al., 2010; Kardol et al., 2010). To predict how soil C and N contents are likely to respond to warming as a future climate scenario under different tree species, it is necessary to establish a new soil–enzyme–C/N model that includes incubation temperature to investigate the effects of warming on soil C and N dynamics.

Current soil organic C models can reproduce changes in C dynamics on various scales under most conditions (Todd-Brown et al., 2013). However, this is not the case in highly variable environments, which may require more models that are

mechanistic and that include enzyme activities (Lawrence et al., 2009; Allison et al., 2010; Li et al., 2010). A few studies have explicitly incorporated enzyme activity into their models and these models have proven to be powerful tools for investigating changes in soil C and N contents in response to warming, as temperature directly affects soil EEA (Shimel and Weintraub, 2003; Lawrence et al., 2009; Davidson et al., 2012). Here, we selected a long-term tree plantation that was

established on a former banana (*Musa acuminata* Colla) farm in subtropical Australia. As these tree plantations were developed from the same soil material, we assume that the current differences in soil properties and litter C/N contents are a 'black box' and are mainly derived from the effects of tree species. We therefore simplified the soil–enzyme–C/N model to consider the effects of both tree species and incubation temperature, although we acknowledge that soil properties such as pH and soil moisture content are important factors influencing soil EEA (Caldwell, 2005; Allison et al., 2010; Kardol et al.,

2010). On the other hand, the enzymatic performance of microbial communities from different tree plantations was explored in a short-term incubation experiment along a laboratory temperature gradient.

We established a new tree species–enzyme–C/N model without considering soil C inputs. The objective of this study was to investigate (1) changes in residual soil C and N contents under different tree species with time and their responses to different temperatures, and (2) differences in residual soil C and N contents between tree species with time in a 78-year-old

forest plantation in subtropical Australia by combining soil EEA assays and a model of the effects of tree species on soil EEA in response to a gradient of incubation temperatures. We hypothesized that long-term tree plantations would change the quality of SOM, thus greatly affecting soil EEA.

## 2 Materials and methods

## 2.1 Experimental site

We selected a 78-year-old forest plantation with different tree species that was established in 1935 on a site that was originally a banana farm. The forest plantation site is located at Cooloola, Tin Can Bay, Southeast Queensland, Australia

($25^{\circ}56'49''$S, $153^{\circ}5'27''$E). The altitude is 43 m above sea level with a mean annual rainfall of 1287 mm. Winter temperatures range from $7^{\circ}$C to $23^{\circ}$C over June to August and summer temperatures range from $18^{\circ}$C to $30^{\circ}$C over December to February (Lu et al., 2012). Four tree species were selected, including an exotic coniferous species (slash pine (*Pinus elliottii* Engelm. var. elliottii)) and two native conifer species (hoop pine (*Araucaria cunninghamii* Ait) and kauri pine (*Agathis robusta* C. Moore)), as well as a *Eucalyptus* species (*Eucalyptus grandis* W Hill ex Maiden). All of them were planted adjacently on a

broad, gently undulating plain with a gentle slope of less than $5^{\circ}$. The plot size of each tree species was 1.087, 0.308, 0.428 and 0.60 ha, respectively (Lu et al., 2012). Four subplots of 10 m × 20 m in each tree plantation were randomly selected for soil sampling, resulting in a total of 16 subplots. The thicknesses of the litter and fermentation layers were 5–6 cm and 1–2 cm for slash pine, respectively, whereas the corresponding values were 4–5 cm and 1–2 cm for the hoop pine and kauri pine plots. The *Eucalyptus* plot had a thicker litter layer of 8–10 cm and a similarly thick fermentation layer of 1–2 cm.

## 2.2 Soil sampling and measurement of soil physiochemical properties

Soil samples were collected in August 2013 using diagonal sampling pattern (i.e., one point at each corner and one in the center of each plot) using a soil auger (8-cm in diameter) at 0–10 cm depth within each plot. The soil cores were immediately mixed thoroughly and kept in a cooler (4 $^o$C). After passing the samples through a 2-mm sieve to remove roots and stones, the soil samples were stored at 4 $^o$C prior to analysis. Part of each fresh sample was stored at 4$^o$C for analysis of soil moisture, pH, and extractable organic C (EOC) and N (EON) (Zhou et al., 2017). The other parts were air-dried and stored at room temperature for soil soluble organic C and N analysis using hot water extraction, and for soil total C and N analysis after being finely ground. Soil moisture content was determined after samples were oven-dried at 105$^o$C overnight. The particle size of these soils was dominated by the sand fraction (~96%). All soil biochemical properties are shown in Table 1.

## 2.3 Measurements of soil enzyme activities

The activity of extracellular enzymes involved in C, N and phosphorus (P) cycling was measured. These enzymes included C acquisition enzymes (β-glucosidase, BG), N acquisition enzymes (*N*-acetylglucosaminidase, NAG; leucine aminopeptidase, LAP) and P acquisition enzymes (acid phosphatases). BG catalyzes one of the steps of cellulose degradation, NAG is involved in chitin and fungal cell wall breakdown, LAP breaks down the polypeptides involved in the mineralization of N from the substrates with polypeptides, and phosphatase is involved in the release of inorganic P. Enzyme activities were assayed spectrophotometrically using *para*-nitrophenol linked substrates (Verchot and Borelli, 2005; Sinsabaugh et al., 2009; Zhou et al., 2013). Briefly, moist field soil (1 g) was suspended in 4 mL of a 0.05 mol/L sodium acetate buffer (pH 6.5 for acid phosphatase, pH 5.0 for all other enzymes). After the substrates (1 mL) were added to Erlenmeyer flasks with the soil solution, the flasks were incubated in the dark at a gradient of temperatures (4, 15, 20, 25, 30 and 37 $^o$C). The duration of incubation depended upon the optimal temperature for each enzyme: 1 h for BG and acid phosphastase, 3 h for NAG and 5 h for LAP at 20, 25, 30 and 37$^o$C; but 1.5 h for BG and 5 h for NAG and 8 h for LAP at 4$^o$C and 15$^o$C. After incubation, 1 M NaOH (4 mL) was added to quench the reaction in the flasks. Enzyme activities were expressed as mg *para*-nitrophenol formed per g dry soil per h.

## 3.4 Modelling and statistical analyses

We assumed that the differences in soil properties and litter C/N contents under different tree species are the results of effects of tree species, and therefore we established a new soil–enzyme–C/N model to consider the effects of both tree

species and incubation temperature without considering other soil properties and litter C inputs derived from tree species. In other words, we considered changes in soil properties and C inputs to be a 'black box' as part of the overall effects of tree species, all of which influenced soil EEA.

We first transformed the enzyme activity data using a natural logarithm. As the enzyme activity data for each plot were not independent along a gradient of temperatures, we needed to consider the interaction of tree species and incubation temperature on soil EEA. So we established a tree species-enzyme model, **Model 1**, to investigate the effects of tree species, incubation temperature and their interaction on soil EEA. **Model 1** is as follows:

$$EEA_{it} = \exp\{\beta_0 + (\beta_1 + X_{1it} \times \beta_{1sp} + X_{2it} \times \beta_{1hp} + X_{3it} \times \beta_{1kp}) \times T + (X_{1it} \times \beta_{2sp} + X_{2it} \times \beta_{2hp} + X_{3it} \times \beta_{2kp}) + \varepsilon_{it}\}$$

$$X_{1it} = \begin{cases} 1 \text{ if the } i^{th} \text{ tree is slash pine,} \\ 0 \text{ otherwise,} \end{cases}$$

$$X_{2it} = \begin{cases} 1 \text{ if the } i^{th} \text{ tree is hoop pine,} \\ 0 \text{ otherwise,} \end{cases}$$

$$X_{3it} = \begin{cases} 1 \text{ if the } i^{th} \text{ tree is kauri pine,} \\ 0 \text{ otherwise,} \end{cases}$$

where $EEA_{it}$ indicates the soil's extracellular enzyme activity; $i$=1, 2, …, 16 is the plot number; $T$ is the temperature (4, 15, 20, 25, 30 and 37°C); $X_{1it}$, $X_{2it}$ and $X_{3it}$ indicate the effects of tree species; $\beta_0$ is a constant; $\beta_1$ is the temperature coefficient; $\beta_{1sp}$, $\beta_{1hp}$, $\beta_{1kp}$, $\beta_{2sp}$, $\beta_{2hp}$ and $\beta_{2kp}$ are the slash pine, hoop pine and kauri pine coefficients, respectively, with temperature and without temperature; and $\varepsilon_{it}$ is the normal distribution with a 0 mean and $\sigma^2_\varepsilon$ variance. We ran all these enzyme activity data in Model 1 and the $F$-test results of the effects of temperature and tree species on soil EEA are presented Table S1.

We found that the interactions between incubation temperature and tree species were not significant on soil EEA (Table S1). Therefore based on the **Model 1**, we further established a simpler model (**Model 2**) without considering their interactions. We also ran a comparison of performance of **Model 1** and **Model 2** using the Akaike information criterion and the Bayesian information criterion (Table S2). Both results show that the performance of **Model 2** is better than **Model 1**, we further updated **Model 1** to create **Model 2**, as shown below:

$$EEA_{it} = \exp\{\beta_0 + \beta_1 \times T + (X_{1it} \times \beta_{2sp} + X_{2it} \times \beta_{2hp} + X_{3it} \times \beta_{2kp}) + \varepsilon_{it}\}. \qquad (2)$$

All the parameters in **Model 2** have been described in **Model 1**. We ran all these enzyme activity data in Model 2 and the $F$-test results are shown in Table S3.

A conventional soil enzyme–C model (**Model 3**) (Schimel and Weintraub, 2003) has been widely used to predict how soil organic C contents change with soil EEA over time.

$$d_{TC}/d_t = -K \times EEA \qquad (3)$$

In this study, for quantitative analysis of the changes in soil total C (TC) contents over time under tree species, we combined **Model 2** with the addition of TC and **Model 3** together to establish a dynamic tree species–enzyme–C model (**Model 4**) as shown below:

$dTC/dt = -K \times EEA,$ where the function of EEA is

$$EEA(T, Tree, TC) = \exp\{\beta_0 + \beta_1 \times T + (X_{1it} \times \beta_{2sp} + X_{2it} \times \beta_{2hp} + X_{3it} \times \beta_{2kp}) + \beta_3 \times TC\},$$

(4) with the boundary conditions $t = time = 0$, $TC = TC_0$, and $TC > 0$.

To get a better understanding these 4 models, we made a simple table to compare the advantages and disadvantages of each model (Table 2). We further got an analytical solution of differential equation for **Model 4** and the equation (Equation 5) is shown below:

$$TC = -1/\beta_3 \times \log\{\beta_3 \times \exp\{\beta_0 + \beta_1 \times T + (X_{1it} \times \beta_{2sp} + X_{2it} \times \beta_{2hp} + X_{3it} \times \beta_{2kp}) + \log(K) \times t + \exp\{-\beta_3 \times TC_0\}\}\},$$

(5)

where $K$ is the unit conversion coefficient when $t = 0$ and $TC = TC_0$.

Similarly, when we consider the relationship between total soil N (TN) and enzyme activities, we get a similar analytical solution of the differential equation (Equation 6) as shown below:

$$TN = -1/\beta_3 \times \log\{\beta_3 \times \exp\{\beta_0 + \beta_1 \times T + (X_{1it} \times \beta_{2sp} + X_{2it} \times \beta_{2hp} + X_{3it} \times \beta_{2kp}) + \log(K) \times t + \exp\{-\beta_3 \times TN_0\}\}\},$$

(6)

where $K$ is the unit conversion coefficient when $t = 0$ and $TN = TN_0$.

Using Equation 5, we calculated the total soil C decomposition over time and compared the residual soil C contents over time among the tree species for different enzyme activities. As the three coniferous tree species showed similar soil C decomposition patterns across time, we only calculated the responses of the residual soil C contents to different temperatures under slash pine and *Eucalyptus*. The model parameters of Equation 5 for different enzyme activities under different tree species are shown in Table S4.

Similarly, using Equation 6, we calculated the total soil N decomposition over time and compared the residual soil N contents among the tree species for different enzyme activities. On the basis of the residual soil C and N contents over time, we finally calculated the ratios of the residual soil C and N contents among the tree species over time. The model parameters of Equation 6 for different enzyme activities under different tree species are shown in Table S5.

**3 Results**

Tree species and temperature significantly affected soil BG, NAG, LAP and acid phosphatase ($P < 0.05$) (Table S1). However, there was no significant effect of the interaction between tree species and temperature on the activity of these enzymes. In general, *Eucalyptus* had 1.01-2.86 times higher soil EEA than the other tree species along a temperature gradient (Fig. 1), followed by native coniferous species (kauri pine and hoop pine), whereas exotic conifer species (slash pine) had the lowest soil enzyme activity, except for NAG enzyme activity (Fig. 1).

The decreasing trends for residual soil C contents over time under different tree species were similar for all enzyme activities (Fig. 2). In general, *Eucalyptus* had the highest soil enzyme activity, followed by slash pine, hoop pine and kauri pine. Notably, *Eucalyptus* had 0.99-1.92 times longer soil C residence time than that of slash pine, 1.56-2.27 times longer than hoop pine, 1.56-2.38 times longer than kauri pine, respectively; slash pine had 1.06-1.77 times longer soil C residence time than the other two coniferous species (Table S6).

Temperature significantly influenced soil C decomposition, indicating that slash pine and *Eucalyptus* had shorter soil C residence times at higher temperatures than at lower temperatures (Fig. 3). At a given temperature, *Eucalyptus* had greater soil C losses, but had higher remaining soil C contents than slash pine (Fig. 3). Interestingly, we noticed that for a certain tree species, the gaps between residual soil C contents with BG at 23°C, 25°C and 27°C increased with time, which may be explained by the canceling effects (absence or strong reduction of response of the enzyme to temperature) of soil EEA (Razavi et al., 2015). Previous findings showed that this phenomena was most pronounced at low substrate concentrations (Razavi et al., 2015), which was consistent with our results in Fig. 3.

We also calculated the ratios of residual soil C contents between *Eucalyptus* and slash pine, between *Eucalyptus* and hoop pine, between *Eucalyptus* and kauri pine, and between slash pine and hoop pine. All of them exhibited consistent and similar patterns and increased with time (Fig. 4).

The decreasing trends for soil N contents under different tree species over time for NAG and LAP showed similar patterns (Fig. 5). *Eucalyptus* had the highest soil EEA, but had 1.39-1.92 times longer decomposition time than that of the other coniferous species. This is because *Eucalyptus* had a relatively flatter slope across time than the other tree species. For NAG, at a given time, *Eucalyptus* had the highest residual soil N contents, followed by (in decreasing order) kauri pine, slash pine and hoop pine. For LAP, at a given time, *Eucalyptus* had the highest residual soil N contents, followed by (in decreasing order) slash pine, hoop pine and kauri pine. The ratios of residual soil N contents between *Eucalyptus* and slash pine for NAG increased with time; similar patterns of the ratios of residual soil N contents between *Eucalyptus* and slash pine over time were seen for LAP (Fig. 6 and Table S7).

## 4 Discussion

In this study we found that long-term forest plantations with different tree species had large differences in the quality of SOM, thus giving significant impacts on soil EEA (Fig. 1, Tables 1 and S1). *Eucalyptus* had the highest soil EEA, which corresponded to higher soil moisture content and total C and N contents than the other coniferous species (Tables 1 and S1). Through using the new tree species–enzyme–C/N model, we clearly show the changes in residual soil C and N contents with time (Figs 2 and 5) and differences in soil C and N residence time (Tables S4 and S5) among different tree species. Soil C and N residence time is very important for predicting soil C and N dynamics (Wallenstein and Weintraub, 2008), as extracellular enzymes play critical role in soil organic C decomposition and N cycling (Todd-Brown et al., 2013). We acknowledged that the actual measurement of C decomposition and C residence time need to use stable isotope techniques.

However, our modeling results provide a relative comparison of soil C and N residence time among different tree species without units (see Figs. 2 and 5). Moreover, our modeling clearly shows that tree species effects on soil C cycling are larger than the effects of the future scenario of temperature increase of $2\,^{\circ}\text{C}$ (see Fig. S1).

## 4.1 Mechanisms for the differences in soil C and N contents between *Eucalyptus* and coniferous trees

We modeled the responses of residual soil C contents to enzyme activity over time (Fig. 2). The results clearly show that *Eucalyptus* had a longer soil C residence time and higher turnover rates (see Table S6) for all enzyme activities than coniferous tree species, even though similar patterns can be seen when soil enzyme activities under slash pine and *Eucalyptus* respond to different temperatures (Fig. 3). The longer soil C residence time under *Eucalyptus* could be attributed to (1) higher initial soil C contents (see Fig. 2), which was supported by the higher forest floor C stocks under *Eucalyptus* than under the other coniferous tree species (Fig. S1), and (2) lower soil pH which can inhibit soil microbial activity (Lu et al., 2012) and increase the specific Acidobacterial group, indicators of soil acidic levels, in soils in this region (Zhou et al., 2017) using high-throughput sequencing. We noticed that there were significantly negative correlations between soil organic C and soil pH ($r = -0.58$, $P < 0.001$).

*Eucalyptus* had a longer mean soil N residence time for LAP and higher soil N turnover rates than the other tree species (Fig. 5 and Table S7). Unlike NAG, which is involved in chitin and fungal cell wall breakdown in the short term, LAP can break down polypeptides in the long term and is involved in the mineralization of N from the substrates with polypeptides (Sinsabaugh et al., 2002; Weand et al., 2010). The longer mean soil N residence time could be another reason for the higher soil C contents seen under *Eucalyptus*, as higher N contents may support plant growth and, in turn, increase soil C stocks.

## 4.2 Mechanisms for differences in soil C and N contents between exotic and native pine species

Exotic coniferous tree species such as slash pine have been widely planted in Eastern Australia (Lu et al., 2012). Slash pine has faster growth rate than the native hoop pine and kauri pine (Maggs, 1985), which was supported by the higher forest floor C stocks (see Fig. S2) and higher soil C stocks (Table 1) under slash pine. Hobbie (2015) synthesized the effects of tree species on soil N cycling and provided four different mechanisms to explain faster plant growth under different tree species. Here, we reported a mechanism for faster growth rate under slash pine. We found that slash pine had a longer soil N residence time than the other coniferous tree species (Figs. 2 and 4), indicating that slash pine has higher residual N contents across time, which may enhance the available N contents in the soil for tree growth. Previous results have shown that slash pine need lower levels of nutrients (N, P and potassium) for its growth, whereas hoop and kauri pines are N-demanding species and are known to accumulate relatively recalcitrant N in forest floor materials (Bubb et al., 1998). On the other hand,

we noticed that slash pine had lower soil pH, which could inhibit microbial decomposition of SOM (Lu et al., 2012), and thus contribute to the longer soil C and N residence time under slash pine than under the other coniferous tree species.

**4.3 Ratios of residual soil C and N contents between tree species**

Global C models usually use a specific parameter to describe soil C decomposition (Todd-Brown et al., 2013). Some of them can simulate microbial decomposition through a new soil biogeochemistry model (Davidson et al., 2012; Steinweg et al., 2012; Wieder et al., 2013). However, the mechanisms of the dynamics of soil C and N contents under different tree species and the mechanisms of the different decomposition times between sites with different soil fertility levels are still unclear (Hobbie, 2015). Here, we found that the more fertile soil seen under *Eucalyptus*, which had higher soil C and N contents, lost more soil C and N, as it had higher C- and N-related enzyme activity. However, as the soil under *Eucalyptus* had a longer soil C and N residence time than the less fertile soil under coniferous tree species with lower soil C and N contents, the differences in the residual soil C and N contents between *Eucalyptus* and coniferous tree species became larger and larger with time (see Figs. 4 and 6). Similarly, the differences in residual soil C and N contents among coniferous tree species became larger and larger with time as well. These results are helpful for understanding soil C and N cycling in tree plantations.

**5 Conclusions**

Both tree species and temperature significantly affected soil EEA. Our modeling analysis clearly shows that though *Eucalyptus* had higher soil EEA activities, it had 0.99-2.38 times longer residual soil C residence time and 1.39-1.92 times longer residual soil N residence time, respectively. Furthermore, the differences in soil C and N contents between *Eucalyptus* and coniferous tree species, as well as between slash pine and hoop pine, became larger and larger with decomposition time, which is in contrast to what we expected. On the other hand, our results help to explain why exotic slash pine can grow faster than the other species studied, as it had 1.22-1.38 times longer residual soil N residence time for LAP than native coniferous tree species like hoop pine and kauri pine. Soil extracellular enzyme assays in combination with statistical modeling, are powerful tools for exploring the mechanisms of soil C and N cycling by different tree species. Our results can provide useful information for local forest management.

**Acknowledgements.** This study was jointly supported by the Griffith University Postdoctoral Fellowship, Griffith University New Researcher Funding and East China Normal University (No. 40500-20101-222011) and National Natural Science Foundation of China (No. 31600406). S. Wang acknowledges the support of the Australian Research Council (DP 150103710).

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

**Table 1.** Soil biochemical properties in 78-year-old forest plantations with different tree species.

| Properties | Slash pine | Hoop pine | Kauri pine | *Eucalyptus* |
|---|---|---|---|---|
| Moisture (%) | 4.26±0.22b | 3.11±0.53b | 3.09±0.67b | 7.69±1.66a |
| pH | 4.58±0.03b | 5.64±0.22a | 6.01±0.23a | 4.49±0.04b |
| Total C (Mg ha$^{-1}$) | 7.36±0.57b | 5.69±0.73b | 5.07±0.75b | 13.88±2.22a |
| Total N (kg ha$^{-1}$) | 232±22b | 245±23b | 239±33b | 462±71a |
| C:N | 31.8±0.7a | 23.1±1.3b | 21.2±0.7b | 29.8±0.6a |
| EOC (mg kg$^{-1}$) | 340±41b | 341±31b | 360±30b | 625±77a |
| EON (mg kg$^{-1}$) | 14.7±2.9b | 18.4±1.9ab | 23.1±1.5a | 22.4±1.8a |
| EOC:EON | 24.17±1.81a | 18.79±1.48b | 15.66±0.87b | 27.64±1.64a |

C, carbon; N, nitrogen; EOC, extractable organic C; EON, extractable organic N

Different letters in the same row indicate significant differences at $P<0.05$ among tree species.

**Table 2.** Description and comparison of the 4 models used in this study.

| | **Model 1** | **Model 2** | **Model 3** | **Model 4** |
|---|---|---|---|---|
| Advantage | Tree species–enzyme model with considering the effects of tree species, incubation temperature and their interactions on soil EEA | Tree species–enzyme model with only considering tree species and incubation temperature on soil EEA, as their interactions were not significant in this study (see Table S1) | Conventional enzyme–C model | Tree species–enzyme–C model by combining **Model 2** and **Model 3** |
| Disadvantage | Without considering the interactions of soil EEA | Without considering the interactions of soil EEA | Without considering the effects of tree species and without considering the interactions of soil EEA | Without considering the interactions of soil EEA |

EEA, extracellular enzyme activities.

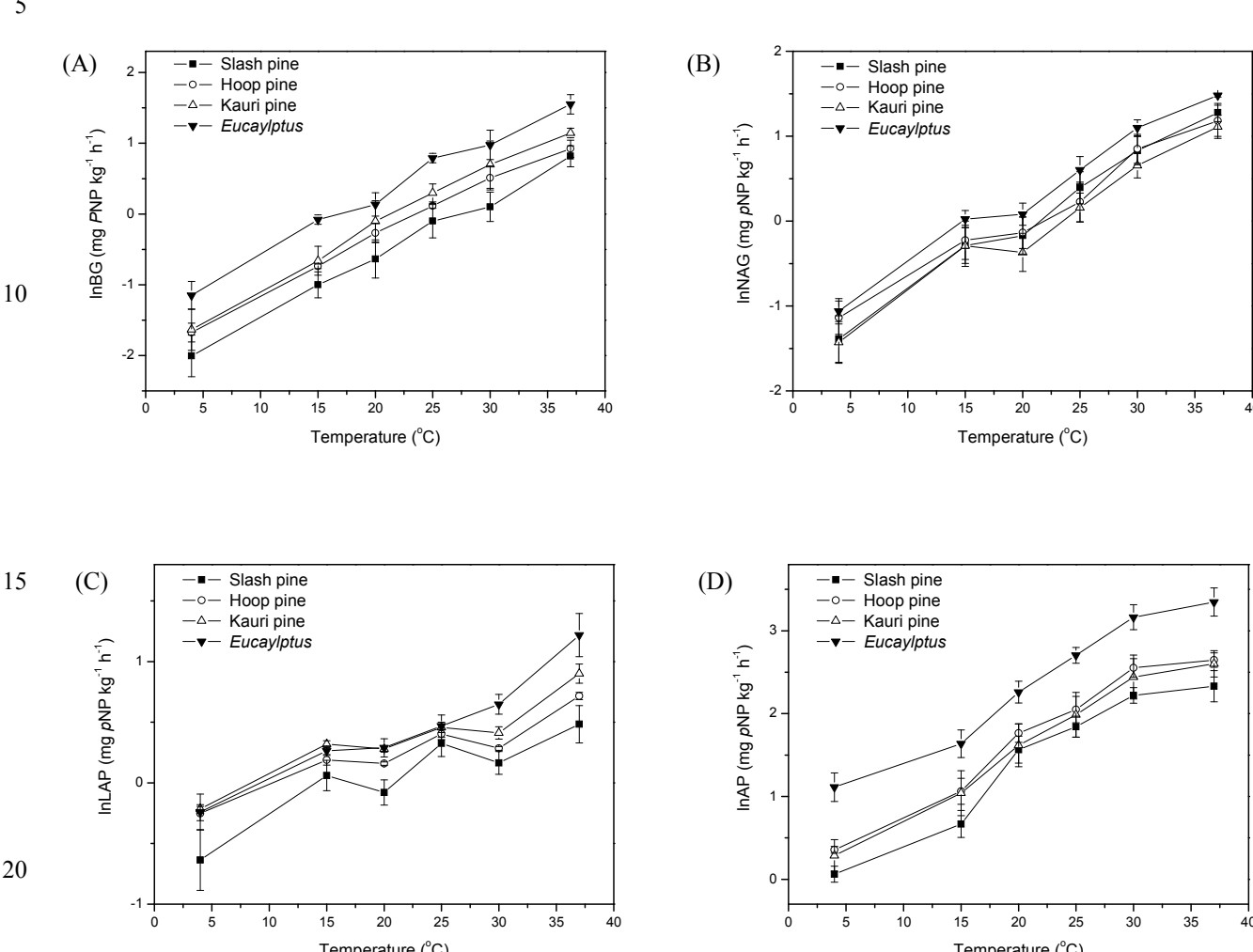

**Figure 1.** Extracellular enzyme activity in the soil along a gradient of temperatures under different tree species. (A) β-glucosidase (BG); (B) *N*-acetylglucosaminidase (NAG); (C) leucine aminopeptidase (LAP); (D) acid phosphatase (AP); *p*NP, *para*-nitrophenol.

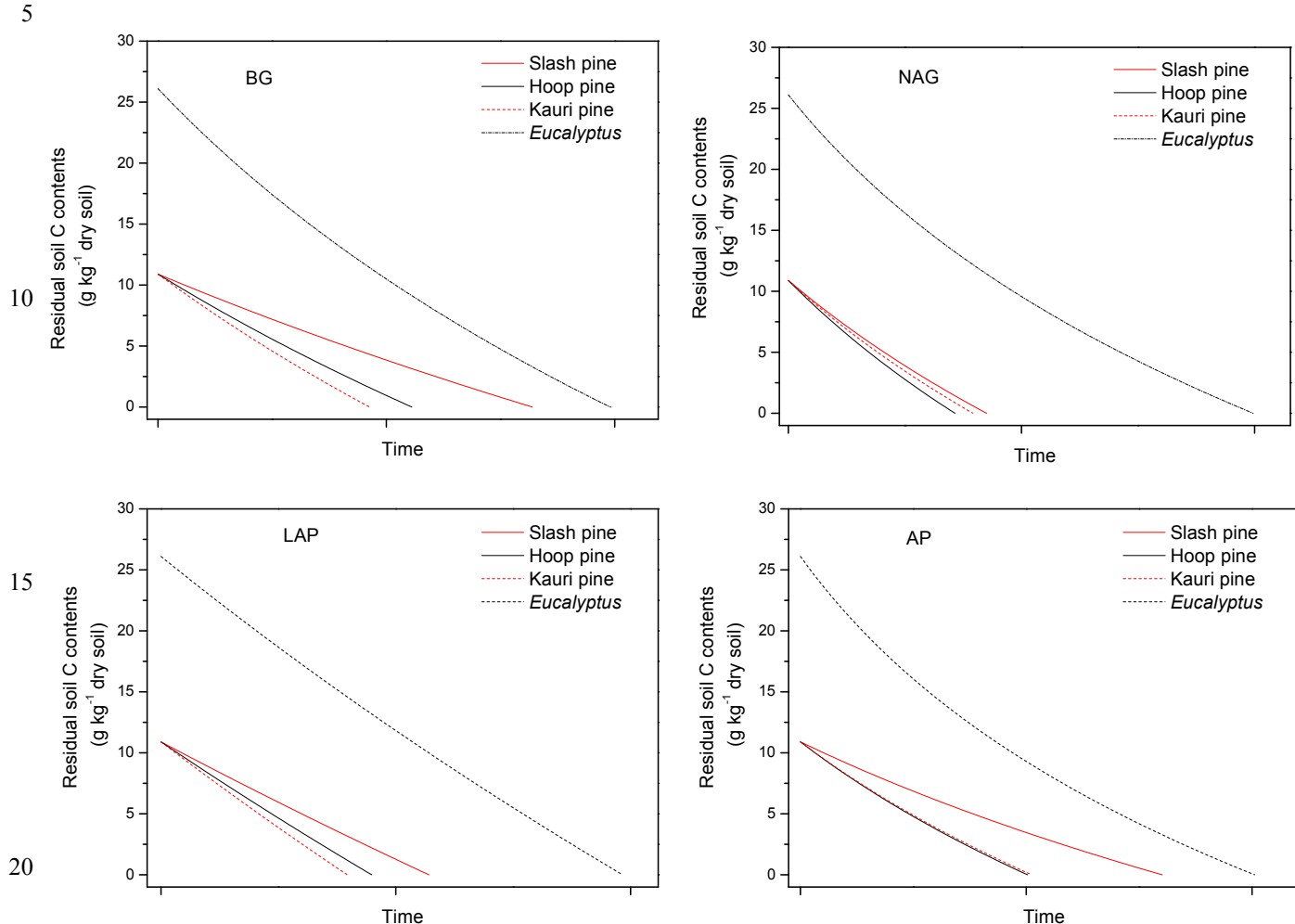

**Figure 2.** Residual soil C contents under different tree species across time for β-glucosidase (BG), *N*-acetylglucosaminidase (NAG), leucine aminopeptidase (LAP) and acid phosphatase (AP) at 25°C. The total soil C decomposition over time was calculated via Equation 5 and the residual soil C contents over time was compared for different enzyme activities among the tree species.

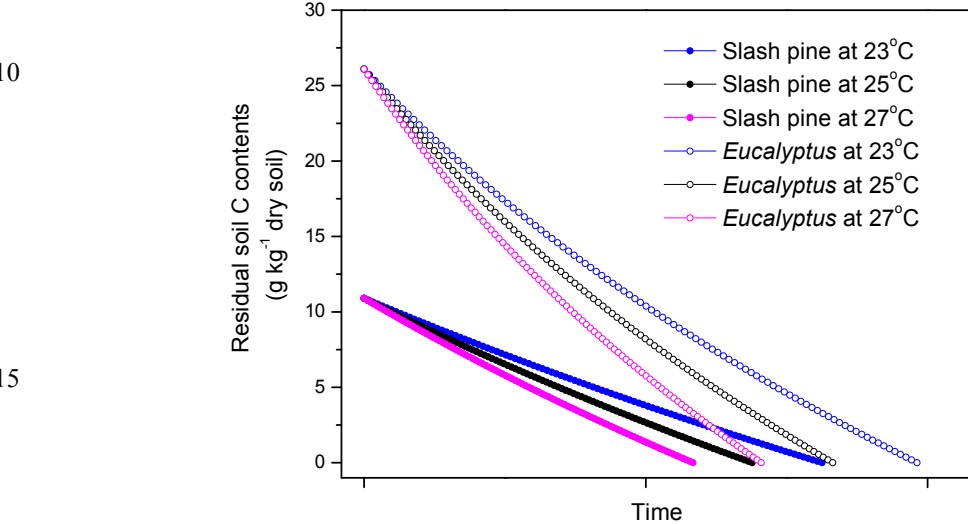

**Figure 3.** Residual soil C contents with β-glucosidase at 23°C, 25°C and 27°C under slash pine and *Eucalyptus* over time.
20   The total soil C decomposition over time was calculated via Equation 5 and the residual soil C contents over time was compared for different enzyme activities among the tree species.

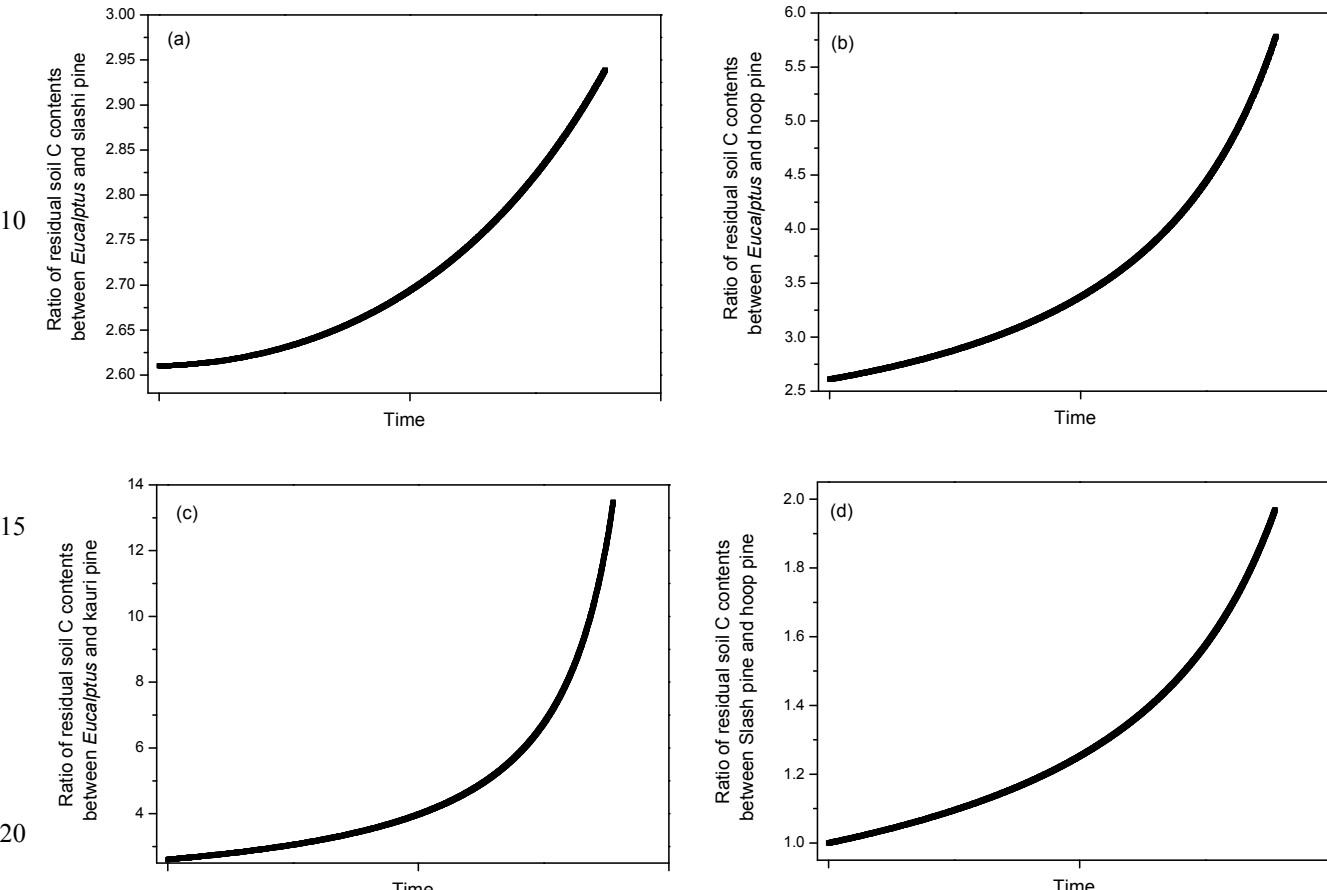

**Figure 4.** Ratios of residual soil C contents between *Eucalyptus* and slash pine (a), between *Eucalyptus* and hoop pine (b), between *Eucalyptus* and kauri pine (c) and between slash pine and hoop pine (d) for β-glucosidase across time.

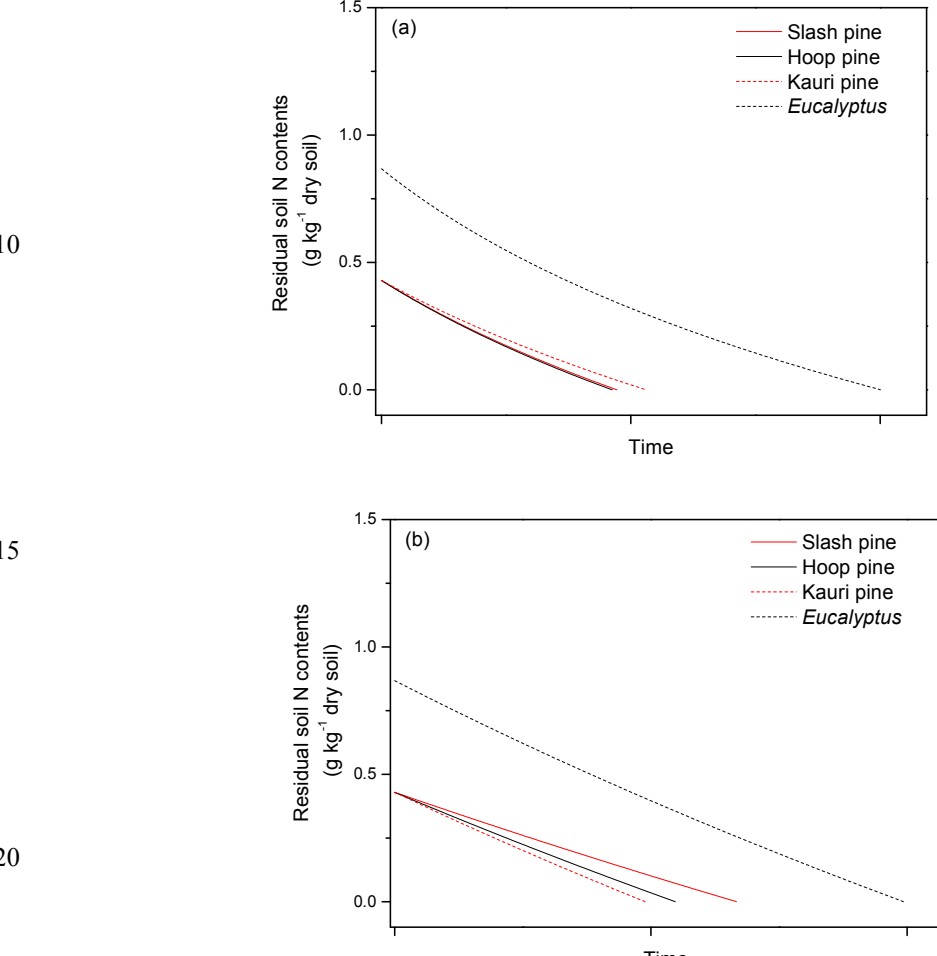

**Figure 5.** Residual soil N contents under different tree species with *N*-acetylglucosaminidase (a) and leucine aminopeptidase (b) across time. The total soil N decomposition over time was calculated via Equation 6 and the residual soil N contents over time was compared for different enzyme activities among the tree species.

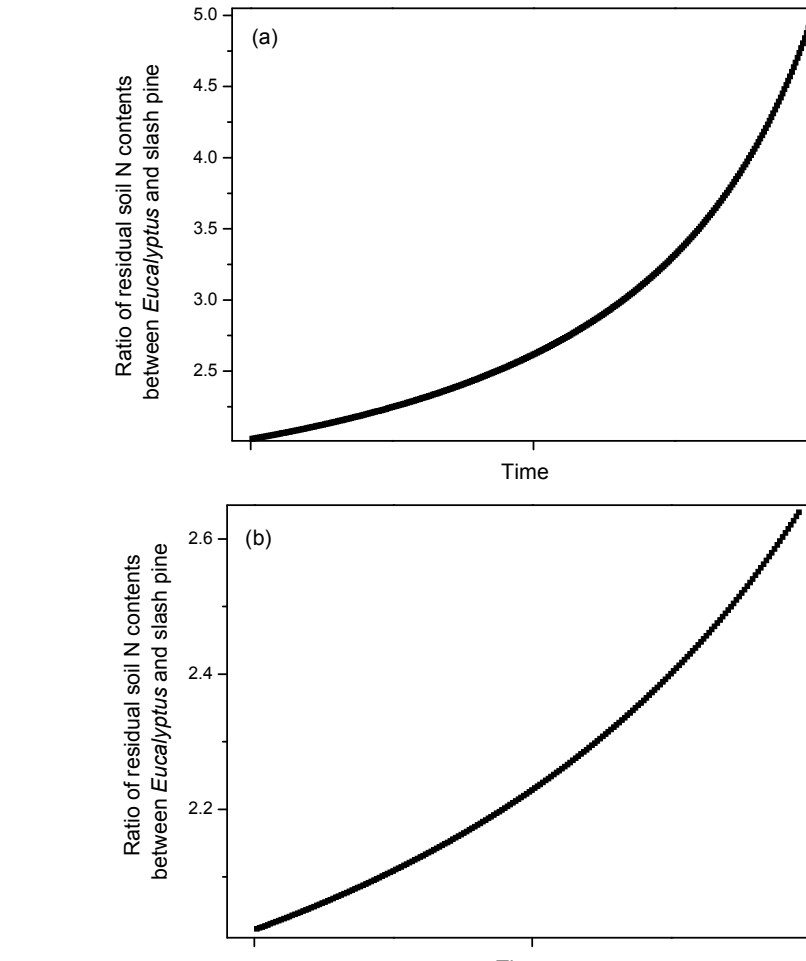

**Figure 6.** Ratios of residual soil N contents between *Eucalyptus* and slash pine for *N*-acetylglucosaminidase (a) and leucine aminopeptidase (b) across time.

Supplementary materials

**Table S1.** *F*-test results showing the effects of tree species and temperature on soil's extracellular enzyme activity in 78-year-old forest plantations with different tree species with Model 1.

|  | BG | NAG | LAP | AP |
|---|---|---|---|---|
| Temperature (T) | 586.39[***] | 493.56[***] | 84.93[***] | 477.71[***] |
| Tree species | 23.44[**] | 5.25[*] | 5.41[*] | 35.46[**] |
| T × tree species | 0.15 | 0.39 | 7.02 | 0.01 |

[*], [**] and [***] indicate significant differences at $P<0.05$, $P<0.01$ and $P<0.001$, respectively.

BG, β-glucosidase; NAG, *N*-acetylglucosaminidase; LAP, leucine aminopeptidase; AP, acid phosphatase

30

**Table S2.** Comparison of the performance of the fitting results between Model 1 and Model 2 using the Akaike information criterion and the Bayesian information criterion.

|  | AIC | | BIC | |
|---|---|---|---|---|
|  | Model 1 | Model 2 | Model 1 | Model 2 |
| BG | 83.25 | 77.73 | 106.33 | 93.12 |
| NAG | 81.50 | 76.73 | 104.58 | 92.12 |
| LAP | 12.73 | 11.57 | 10.35 | 3.82 |
| AP | 80.45 | 74.47 | 103.53 | 89.86 |

15 BG, β-glucosidase; NAG, *N*-acetylglucosaminidase; LAP, leucine aminopeptidase; AP, acid phosphatase; AIC, Akaike information criterion; BIC, Bayesian information criterion

**Table S3.** *F*-test results showing the effects of tree species and temperature on soil's extracellular enzyme activity with Model 2.

| | BG | NAG | LAP | AP |
|---|---|---|---|---|
| Temperature | 603.35[***] | 533.38[***] | 134.92[***] | 502.63[***] |
| Tree species | 24.12[**] | 6.16[*] | 7.37[**] | 37.91[**] |

[*], [**] and [***] indicate significant differences at $P<0.05$, $P<0.01$ and $P<0.001$, respectively.

BG, β-glucosidase; NAG, *N*-acetylglucosaminidase; LAP, leucine aminopeptidase; AP, acid phosphatase

**Table S4.** Model parameters of Equation 5 for residual soil C contents under different tree species across time with different extracellular enzyme activities.

|  | BG | NAG | LAP | AP |
|---|---|---|---|---|
| $\beta_0$ | -2.19785309 | -2.39483244 | -0.55407280 | -0.50758993 |
| $\beta_1$ | 0.08273358 | 0.07592116 | 0.03148246 | 0.07480559 |
| $\beta_{2sp}$ | -0.48260712 | 0.25997095 | -0.24294259 | -0.33588831 |
| $\beta_{2hp}$ | -0.09399809 | 0.43360852 | -0.00285819 | 0.12975114 |
| $\beta_{2kp}$ | 0.08828864 | 0.33102009 | 0.12017906 | 0.11112443 |
| $\beta_3$ | 0.02919402 | 0.04245054 | 0.01179556 | 0.04770020 |

BG, β-glucosidase; NAG, *N*-acetylglucosaminidase; LAP, leucine aminopeptidase; AP, acid phosphatase

**Table S5.** Model parameters of Equation 6 for residual soil N contents under different tree species across time with different extra-cellular enzyme activities.

|  | NAG | LAP |
| --- | --- | --- |
| $\beta_0$ | -2.39203701 | -0.56717981 |
| $\beta_1$ | 0.07592116 | 0.03148246 |
| $\beta_{2sp}$ | 0.28903302 | -0.22794473 |
| $\beta_{2hp}$ | 0.30922792 | -0.03046445 |
| $\beta_{2kp}$ | 0.17212690 | 0.08319450 |
| $\beta_3$ | 1.27357366 | 0.36988125 |

NAG, *N*-acetylglucosaminidase; LAP, leucine aminopeptidase

**Table S6.** Half-residence time and C turnover rate of residual soil C contents under different tree species with different extra-cellular enzyme activities. The total soil C decomposition over time was calculated via Equation 5. We set the half-residence time under *Eucalyptus* for different enzyme activities as $t_i$($i$=1, 2, 3, 4) and compared it with other half-residence times under coniferous tree species. The C turnover rate for each enzyme was calculated from half the residual soil C contents divided by the half -residence times under different tree species.

| | Half-residence time | | | | C turnover rate | | | |
|---|---|---|---|---|---|---|---|---|
| | BG | NAG | LAP | AP | BG | NAG | LAP | AP |
| Slash pine | $0.94t_1$ | $0.52t_2$ | $0.61t_3$ | $1.01t_4$ | $5.80/t_1$ | $10.48/t_2$ | $8.93/t_3$ | $5.40/t_4$ |
| Hoop pine | $0.64t_1$ | $0.44t_2$ | $0.48t_3$ | $0.63t_4$ | $8.52/t_1$ | $12.39/t_2$ | $11.35/t_3$ | $8.65/t_4$ |
| Kauri pine | $0.53t_1$ | $0.49t_2$ | $0.42t_3$ | $0.64t_4$ | $10.28/t_1$ | $11.12/t_2$ | $12.98/t_3$ | $8.52/t_4$ |
| *Eucalyptus* | $t_1$ | $t_2$ | $t_3$ | $t_4$ | $13.05/t_1$ | $13.05/t_2$ | $13.05/t_3$ | $13.05/t_4$ |

BG, β-glucosidase; NAG, *N*-acetylglucosaminidase; LAP, leucine aminopeptidase; AP, acid phosphatase

**Table S7.** Half-residence time and N turnover rate of residual soil N contents under different tree species with different extracellular enzyme activities. The total soil N decomposition over time was calculated via Equation 6. We set the half-residence time under *Eucalyptus* for different enzyme activities as $t_i(i=1, 2)$ and compared it with other half-residence times under coniferous tree species. The N turnover rate for each enzyme was calculated from half the residual soil N contents divided by the half -residence times under different tree species.

|  | Half residence time | | N turnover rate | |
|---|---|---|---|---|
|  | NAG | LAP | NAG | LAP |
| Slash pine | $0.56t_1$ | $0.72t_2$ | $0.38/t_1$ | $0.30/t_2$ |
| Hoop pine | $0.55t_1$ | $0.59t_2$ | $0.39/t_1$ | $0.36/t_2$ |
| Kauri pine | $0.63 t_1$ | $0.52t_2$ | $0.34/t_1$ | $0.41/t_2$ |
| *Eucalyptus* | $t_1$ | $t_2$ | $0.43/t_1$ | $0.43/t_2$ |

NAG, *N*-acetylglucosaminidase; LAP, leucine aminopeptidase

30

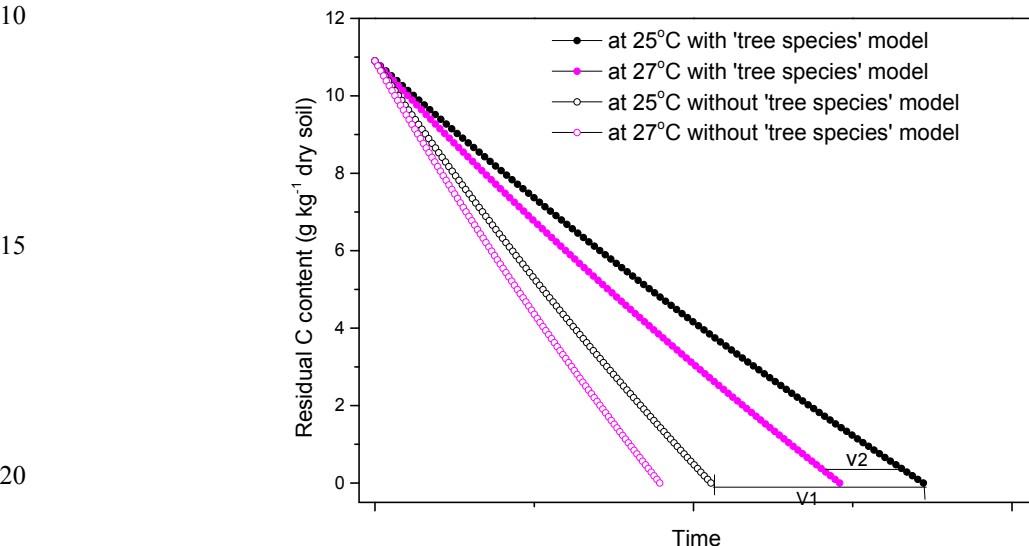

**Figure S1** A new model without considering the effect of tree species showing the changes in soil residual C contents at 25$^o$C and at 27$^o$C with and without the effects of tree species. V1 indicates the differences in residual C contents with and
25 without the tree species model, and V2 indicates the differences in residual C contents at 25$^o$C and at 27$^o$C with the tree species model.

30

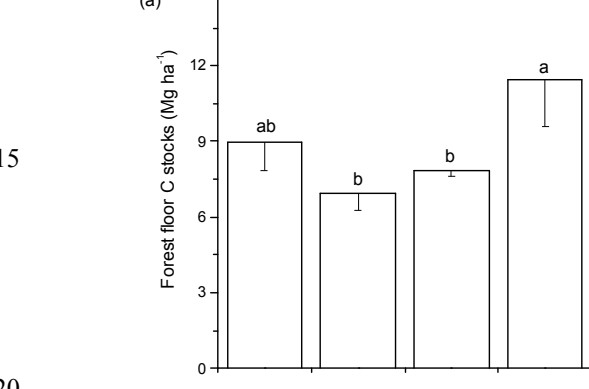 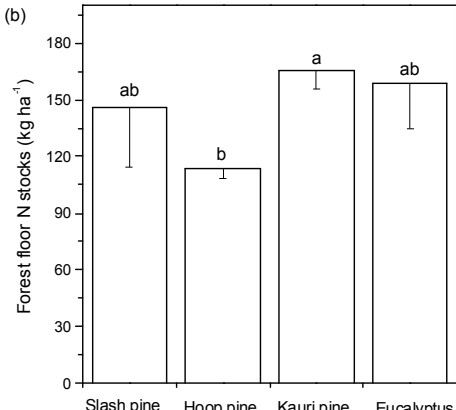

**Figure S2** Differences in forest floor C (a) and N (b) stocks under different tree species. Different letters significant differences at *P*<0.05 among the treatments.