# Peer review of "Modeling the effects of tree species and incubation temperature on soil's extracellular enzyme activity in 78-year-old tree plantations"

_Biogeosciences, 2017_

## Referee Comment (RC1) · Anonymous Referee #1 · 22 Jun 2017

General comments:

This study modelled the effects of tree species and temperature on soil extracellular enzyme activities in a 78 year old plantation in southeast Queensland of Australia. This is an interesting topic and the MS is generally well written. However, there are some major limitations which I think need to be addressed before accepted for publication.

My biggest concern is that some writing and terms did not reflect the actual design of the study. For instance, the authors concluded that Eucalyptus had highest EEA, long C residence time and lowest C turnover rate. This is contradicting. Higher enzyme activities mean quicker C turnover. The actual C decomposition and stabilisation, and

C residence time in soil cannot be measured without the use of stable isotopes. While results show clear differences between different tree species, it is arbitrary to make such conclusions. Therefore the conclusions need to be more relevant to the design and topic.

Also, the title and many parts of the introduction make the readers believe this study is to investigate the effect of warming on soil EEA. Actually, the incubation under the temperature gradient is just a methodology study of enzyme activities incubation with substrates at different temperatures. It's not an actual experiment testing climate warming on soil microbial properties. The readers can easily be confused.

The introduction did not clearly identify the knowledge gaps in this area. Was not no study on tree species on EEA? If any what did they tell us and what the current research gap is? Also the authors only mentioned very briefly in the introduction temperature effects. More work needs to be done.

The DISCUSSION is the weakest part of this MS. Clearly the results and relevant implications have not been explicitly discussed.

Specific comments:

Lines 67-72, the logic is reversed. Better say climate warming would increase SOM decomposition, and, in return, C losses from SOM decomposition would have feedbacks on atmospheric $CO_2$ concentration and global temperature.

Lines 92-93, Vague, what's the point of this sentence?

Lines 94-96, Again this sentence is not clear. Why do microbes produce less EEA under warmer temperatures?

Lines 107-109, what are these models about? If incorporated EEA, what did they tell us and what are their limitations so you need to develop a new model?

Lines 126-128, as the hypotheses, please be specific. Everyone know that EEA would

be different under different tree species and temperature.

Line 160, where is the result for MBC and MBN?

Lines 304-305, yes and obvious, not surprising.

Lines 308-317, all textbook sentences, repetitive.

Lines 323-324, Eucalyptus has highest EEA but lowest C turnover rate? Sounds contradicting and hard to understand. I think it has quick turnover, and the reason it's still building soil C is because of its massive input.

Line 328, the reason for low pH could be a direct result of SOM decomposition caused acidification, which means quicker turnover.

Lines 343-350, all RESULTS sentences, no DISCUSSION.

---

## Author Comment (AC1) · 17 Jul 2017

[revised manuscript text omitted]
 extracellular enzyme activities in a 78 year old plantation in southeast Queensland of Australia. This is an interesting topic and the MS is generally well written. However, there are some major limitations which I think need to be addressed before accepted for publication. My biggest concern is that some writing and terms did not reflect the actual design of the study. For instance, the authors concluded that Eucalyptus had highest EEA, long C

10   residence time and lowest C turnover rate. This is contradicting. Higher enzyme activities mean quicker C turnover. The actual C decomposition and stabilisation, and C residence time in soil cannot be measured without the use of stable isotopes. While results show clear differences between different tree species, it is arbitrary to make such conclusions. Therefore the conclusions need to be more relevant to the design and topic.

15   R: Thanks. We re-calculated the soil C turnover rate in Tables S4 and S5 as shown below. You are right that *Eucalyptus* has the highest soil C and N turnover rates as it has the highest initial soil C and N contents. We acknowledged that we did not consider differences in initial soil C and N contents before. However, the differences in C turnover rates of the three coniferous tree species did not change. Secondly, We acknowledged that the actual measurement of C decomposition and C residence time

20   need to use stable isotope techniques. However, our modeling results provide a relative comparison of soil C and N residence time among different tree species without units (see Figs. 2 and 5). Lines 24-26 Page 7

**Table S4.** Half-residence time and C turnover rate of residual soil C contents under different tree species with different

25   extra-cellular enzyme activities. The total soil C decomposition over time was calculated via Equation 5. We set the half-residence time under *Eucalyptus* for different enzyme activities as $t_i$($i$=1, 2, 3, 4) and compared it with other half-residence times under coniferous tree species. The C turnover rate for each enzyme was calculated from half the residual soil C contents divided by the half-residence times under different tree species.

| | Half-residence time | | | | C turnover rate | | | |
|---|---|---|---|---|---|---|---|---|
| | BG | NAG | LAP | AP | BG | NAG | LAP | AP |
| Slash pine | $0.94t_1$ | $0.52t_2$ | $0.61t_3$ | $1.01t_4$ | $5.80/t_1$ | $10.48/t_2$ | $8.93/t_3$ | $5.40/t_4$ |
| Hoop pine | $0.64t_1$ | $0.44t_2$ | $0.48t_3$ | $0.63t_4$ | $8.52/t_1$ | $12.39/t_2$ | $11.35/t_3$ | $8.65/t_4$ |
| Kauri pine | $0.53t_1$ | $0.49t_2$ | $0.42t_3$ | $0.64t_4$ | $10.28/t_1$ | $11.12/t_2$ | $12.98/t_3$ | $8.52/t_4$ |
| *Eucalyptus* | $t_1$ | $t_2$ | $t_3$ | $t_4$ | $13.05/t_1$ | $13.05/t_2$ | $13.05/t_3$ | $13.05/t_4$ |

BG, β-glucosidase; NAG, *N*-acetylglucosaminidase; LAP, leucine aminopeptidase; AP, acid phosphatase

**Table S5.** Half-residence time and N turnover rate of residual soil N contents under different tree species with different extracellular enzyme activities. The total soil N decomposition over time was calculated via Equation 6. We set the half-residence time under *Eucalyptus* for different enzyme activities as $t_i(i=1, 2)$ and compared it with other half-residence times under coniferous tree species. Nitrogen turnover rate for each enzyme was calculated from half the residual soil N contents divided by the half-residence times under different tree species.

| | Half residence time | | N turnover rate | |
|---|---|---|---|---|
| | NAG | LAP | NAG | LAP |
| Slash pine | $0.56t_1$ | $0.72t_2$ | $0.38/t_1$ | $0.30/t_2$ |
| Hoop pine | $0.55t_1$ | $0.59t_2$ | $0.39/t_1$ | $0.36/t_2$ |
| Kauri pine | $0.63\ t_1$ | $0.52t_2$ | $0.34/t_1$ | $0.41/t_2$ |
| *Eucalyptus* | $t_1$ | $t_2$ | $0.43/t_1$ | $0.43/t_2$ |

NAG, *N*-acetylglucosaminidase; LAP, leucine aminopeptidase
Lines 12-16 Page 24
Lines 12-16 Page25

Also, the title and many parts of the introduction make the readers believe this study is to investigate the effect of warming on soil EEA. Actually, the incubation under the temperature gradient is just a methodology study of enzyme activities incubation with substrates at different temperatures. It's not an actual experiment testing climate warming on soil microbial proper ties. The readers can easily be confused.
R: Thanks. We have changed the title into 'incubation temperature'. We want to establish a new soil–enzyme–C/N model that includes incubation temperature to investigate the effects of warming on soil C and N dynamics to predict how soil C and N contents are likely to respond to warming as a future climate scenario under different tree species.
Line 1 Page 1

The introduction did not clearly identify the knowledge gaps in this area. Was not no study on tree species on EEA? If any what did they tell us and what the current research gap is? Also the authors only mentioned very briefly in the introduction temperature effects. More work needs to be done.
R: This part has been improved.
We assume that the current differences in soil properties and litter C/N contents are a 'black box' and are mainly derived from the effects of tree species. We therefore simplified soil–enzyme–C/N model to consider the effects of both tree species and incubation temperature.
Lines 11-16 Page 3

The DISCUSSION is the weakest par t of this MS. Clearly the results and relevant implications have not been explicitly discussed.

R: This has been improved.

In this study we found that long-term forest plantations with different tree species had large differences in the quality of SOM, thus giving significant impacts on soil EEA (Fig. 1, Tables 1 and 2). *Eucalyptus* had the highest soil EEA, which corresponded to higher soil moisture content and total C and N contents than the other coniferous species (Tables 1 and 2). Through using the new tree species–enzyme–C/N model, we clearly show the changes in residual soil C and N contents with time (Figs 2 and 5) and differences in soil C and N residence time (Tables S4 and S5) among different tree species. Soil C and N residence time is very important for predicting soil C and N dynamics (Wallenstein and Weintraub, 2008), as extracellular enzymes play critical role in soil organic C decomposition and N cycling (Todd-Brown et al., 2013). We acknowledged that the actual measurement of C decomposition and C residence time need to use stable isotope techniques. However, our modeling results provide a relative comparison of soil C and N residence time among different tree species without units (see Figs. 2 and 5). Moreover, our modeling clearly shows that tree species effects on soil C cycling are larger than the effects of the future scenario of temperature increase of 2 $^{\circ}$C (see Fig. S1).

Lines 18-28 Page 7

Exotic coniferous tree species such as slash pine have been widely planted in Eastern Australia (Lu et al., 2012). Slash pine has faster growth rate than the native hoop pine and kauri pine (Maggs, 1985), which was supported by the higher forest floor C stocks (see Fig. S2) and higher soil C stocks (Table 1) under slash pine. Hobbie (2015) synthesized the effects of tree species on soil N cycling and provided four different mechanisms to explain faster plant growth under different tree species. Here, we reported a mechanism for faster growth rate under slash pine. We found that slash pine had a longer soil N residence time than the other coniferous tree species (Figs. 2 and 4), indicating that slash pine has higher residual N contents across time, which may enhance the available N contents in the soil for tree growth. Previous results have shown that slash pine need lower levels of nutrients (N, P and potassium) for its growth, whereas hoop and kauri pines are N-demanding species and are known to accumulate relatively recalcitrant N in forest floor materials (Bubb et al., 1998). On the other hand, we noticed that slash pine had lower soil pH, which could inhibit microbial decomposition of SOM (Lu et al., 2012), and thus contribute to the longer soil C and N residence time under slash pine than under the other coniferous tree species.

Lines 16-26 Page 8

Specific comments:

Lines 67-72, the logic is reversed. Better say climate warming would increase SOM decomposition, and, in return, C losses from SOM decomposition would have feedbacks on atmospheric CO2 concentration and global temperature.

R: This has been revised.

In the context of climate warming, minor losses of C via decomposition of soil organic matter (SOM) can cause positive feedback to atmospheric $CO_2$ concentrations and global temperature (IPCC, 2013), resulting in increase in plant growth and decomposition of SOM (Davidson and Janssens, 2006; Wu et al., 2011), which can profoundly alter soil C and nitrogen (N) cycling (Luo, 2007).
Lines 6-9 Page 2

Lines 92-93, Vague, what's the point of this sentence?
Lines 94-96, Again this sentence is not clear. Why do microbes produce less EEA under warmer temperatures?
R: This has been revised.
Previous work has shown that in a variety of ecosystems, the enzymatic activities associated with decomposition differ, depending upon the quality of SOM such as soil C:N ratios (Sinsabaugh et al., 2002). Forest plantations with different tree species have been reported to have large differences in the quantity and quality of SOM, thus greatly influencing soil EEA (Lovett et al., 2004; Lu et al., 2012; Hobbie, 2015). Generally, as soil EEA increases with increasing temperature (Koch et al., 2007), to get a certain amount of substrate via decomposition of SOM, microbes in warmer soils need to produce fewer extracellular enzymes that are involved in C and nutrient cycling (Allison, 2005).
Lines 22-27 Page 2

Lines 107-109, what are these models about? If incorporated EEA, what did they tell us and what are their limitations so you need to develop a new model?
R: This has been revised.
Current soil organic C models can reproduce changes in C dynamics on various scales under most conditions (Todd-Brown et al., 2013). However, this is not the case in highly variable environments, which may require more models that are mechanistic and that include enzyme activities (Lawrence et al., 2009; Allison et al., 2010; Li et al., 2010). A few studies have explicitly incorporated enzyme activity into their models and these models have proven to be powerful tools for investigating changes in soil C and N contents in response to warming, as temperature directly affects soil EEA.
Lines 32-34 Page 2
Lines 1-2 Page 3

Lines 126-128, as the hypotheses, please be specific. Everyone know that EEA would be different under different tree species and temperature.
R: This has been revised.
We established a new tree species–enzyme–C/N model without considering soil C inputs. The objective of this study was to investigate (1) changes in residual soil C and N contents under different tree species with time and their responses to different temperatures, and (2) differences in residual soil C and N contents between tree species with time in a 78-year-old forest plantation in subtropical

Australia by combining soil EEA assays and a model of the effects of tree species on soil EEA in response to a gradient of incubation temperatures. We hypothesized that long-term tree plantations would change the quality of SOM, thus greatly affecting soil EEA.
Lines 11-16 Page 3

Line 160, where is the result for MBC and MBN?
R: This has been revised.
extractable organic C (EOC) and N (EON)
Line 7 Page 4

Lines 304-305, yes and obvious, not surprising.
R: This paragraph has been revised.
Lines 18-28 Page 7

Lines 308-317, all textbook sentences, repetitive.
R: This has been revised.
In this study we found that long-term forest plantations with different tree species had large differences in the quality of SOM, thus giving significant impacts on soil EEA (Fig. 1, Tables 1 and 2). *Eucalyptus* had the highest soil EEA, which corresponded to higher soil moisture content and total C and N contents than the other coniferous species (Tables 1 and 2). Through using the new tree species–enzyme–C/N model, we clearly show the changes in residual soil C and N contents with time (Figs 2 and 5) and differences in soil C and N residence time (Tables S4 and S5) among different tree species. Soil C and N residence time is very important for predicting soil C and N dynamics (Wallenstein and Weintraub, 2008), as extracellular enzymes play critical role in soil organic C decomposition and N cycling (Todd-Brown et al., 2013). We acknowledged that the actual measurement of C decomposition and C residence time need to use stable isotope techniques. However, our modeling results provide a relative comparison of soil C and N residence time among different tree species without units (see Figs. 2 and 5). Moreover, our modeling clearly shows that tree species effects on soil C cycling are larger than the effects of the future scenario of temperature increase of 2 $^{\circ}$C (see Fig. S1).
Lines 18-28 Page 7

Lines 323-324, Eucalyptus has highest EEA but lowest C turnover rate? Sounds contradicting and hard to understand. I think it has quick turnover, and the reason it's still building soil C is because of its massive input.
R: Thanks. We re-calculated the soil C turnover rate in Tables S4 and S5 as shown below. You are right that *Eucalyptus* has the highest soil C and N turnover rates as it has the highest initial soil C and N contents. We acknowledged that we did not consider differences in initial soil C and N contents before. However, the differences in C turnover rate of the three coniferous tree species did not change.
Lines 33 Page 7

Line 328, the reason for low pH could be a direct result of SOM decomposition caused acidification, which means quicker turnover.

R: This has been revised.

5 The longer soil C residence time under *Eucalyptus* could be attributed to (1) higher initial soil C contents (see Fig. 2), which was supported by the higher forest floor C stocks under *Eucalyptus* than under the other coniferous tree species (Fig. S1), and (2) lower soil pH which can inhibit soil microbial activity (Lu et al., 2012) and increase the specific Acidobacterial group, indicators of soil acidic levels, in soils in this region (Zhou et al., 2017) using high-throughput sequencing. We noticed that there were

10 significantly negative correlations between soil organic C and soil pH ($r = -0.58$, $P < 0.001$).
Lines 1-6 Page 8

Lines 343-350, all RESULTS sentences, no DISCUSSION

R: This paragraph has been revised.

15 Exotic coniferous tree species such as slash pine have been widely planted in Eastern Australia (Lu et al., 2012). Slash pine has faster growth rate than the native hoop pine and kauri pine (Maggs, 1985), which was supported by the higher forest floor C stocks (see Fig. S2) and higher soil C stocks (Table 1) under slash pine. Hobbie (2015) synthesized the effects of tree species on soil N cycling and provided four different mechanisms to explain faster plant growth under different tree species. Here, we reported

20 a mechanism for faster growth rate under slash pine. We found that slash pine had a longer soil N residence time than the other coniferous tree species (Figs. 2 and 4), indicating that slash pine has higher residual N contents across time, which may enhance the available N contents in the soil for tree growth. Previous results have shown that slash pine need lower levels of nutrients (N, P and potassium) for its growth, whereas hoop and kauri pines are N-demanding species and are known to accumulate

25 relatively recalcitrant N in forest floor materials (Bubb et al., 1998). On the other hand, we noticed that slash pine had lower soil pH, which could inhibit microbial decomposition of SOM (Lu et al., 2012), and thus contribute to the longer soil C and N residence time under slash pine than under the other coniferous tree species.
Lines 16-26 Page 8

---

## Referee Comment (RC2) · Anonymous Referee #3 · 25 Sep 2017

I'm not a specialist of the extracellular enzyme activity, but I guess that soil water content is also important factor for soil's extracellular enzyme activity, but authors did not make mention of that. Why? I think that it is very serious problem in this manuscript. Please complete a number of the y-axis of all figures in Figure 4 and Figure 6. For example, Figure 6a and 6b are very similar shape, but if these y-axis were same, the shapes would be different. Additionally I cannot understand the importance of results of Figure 4. Authors calculate the residual soil C contents between two species, but I cannot understand the method of this calculation.

---

## Author Comment (AC2) · 2 Oct 2017

[revised manuscript text omitted]

I'm not a specialist of the extracellular enzyme activity, but I guess that soil water content is also important factor for soil's extracellular enzyme activity, but authors did not make mention of that. Why? I think that it is very serious problem in this manuscript.

R: Thanks for your suggestions.

We acknowledge that, similar to other soil properties such as soil C contents and soil pH etc, soil moisture content plays an important role in regulating soil extracellular enzyme activities (EEA).

In this study, we selected a long-term tree plantation that was established on a former banana (*Musa acuminata Colla*) farm in subtropical Australia. As these tree plantations were developed from the same soil material, we assume that the current differences in soil properties including soil moisture contents are a 'black box' and are mainly derived from the effects of tree species. We therefore simplified the soil–enzyme–C/N model to consider the effects of both tree species and incubation temperature, although we acknowledge that soil properties such as soil moisture content are important factors influencing soil EEA (Caldwell, 2005; Allison et al., 2010; Kardol et al., 2010). (see Page 3 Lines 3-16)

In this manuscript we mainly focus on changes in residual soil C and N contents under different tree species with time and their responses to different temperatures. We found that *Eucalyptus* had larger soil C losses but had longer soil C residence time than the coniferous tree species over time. The differences in the residual soil C and N contents between *Eucalyptus* and coniferous tree species, as well as between slash pine (*Pinus elliottii* Engelm. var. *elliottii*) and hoop pine (*Araucaria cunninghamii* Ait), become larger and larger over time.

To our knowledge, it is the first time to clearly show that when soils with different C contents are subject to climate warming, high C soil has more C losses but differences in residual C between high C and low C soils become larger and larger. This is similar to economical phenomena, i.e., when a society is subject to financial crisis, the rich lose more money but fortune gap between the rich and the poor become larger and larger.

Please complete a number of the y-axis of all figures in Figure 4 and Figure 6. For example, Figure 6a and 6b are very similar shape, but if these y-axis were same, the shapes would be different.

R: We have changed the y-axis in Fig. 6a and Fig. 6b to the same scale range as shown below. The patterns of ratios of residual N contents between tree species is similar between these figures.

[Figure]

Figure 6. Ratios of residual soil N contents between *Eucalyptus* and slash pine for N-acetylglucosaminidase (a) and leucine aminopeptidase (b) across time.

Page 20 Lines 23-24

Additionally I cannot understand the importance of results of Figure 4.

R: Fig. 4 shows that the differences in the residual soil C contents between *Eucalyptus* and coniferous tree species, as well as between slash pine (*Pinus elliottii* Engelm. var. *elliottii*) and hoop pine (*Araucaria cunninghamii* Ait), become larger and larger over time.

5    To our knowledge, it is the first time to clearly show that when soils with different C contents are subject to climate warming, high C soil has more C losses but differences in residual C between high C and low C soils become larger and larger. This is similar to economical phenomena, i.e., when a society is subject to financial crisis, the rich lose more money but fortune gap between the rich and the poor become larger and larger.

Authors calculate the residual soil C contents between two species, but I cannot understand the method of this calculation.

R: Residual C contents over time was calculated based on Equation 5.

The analytical solution of differential equations (5) is shown below:

15    $TC = -1/\beta_3 \times \log\{\beta_3 \times \exp\{\beta_0 + \beta_1 \times T + (X_{1it} \times \beta_{2sp} + X_{2it} \times \beta_{2hp} + X_{3it} \times \beta_{2kp}) + \log(K) \times t + \exp\{-\beta_3 \times TC_0\}\}\}$, (5)

where $K$ is the unit conversion coefficient when $t = 0$ and $TC = TC_0$.

Page 6 Lines 3-5

These model parameters have been given in supplementary materials as shown below.

20  **Table S2.** Model parameters of Equation 5 for residual soil C contents under different tree species across time with different extracellular enzyme activities.

|  | BG | NAG | LAP | AP |
|---|---|---|---|---|
| $\beta_0$ | -2.19785309 | -2.39483244 | -0.55407280 | -0.50758993 |
| $\beta_1$ | 0.08273358 | 0.07592116 | 0.03148246 | 0.07480559 |
| $\beta_{2sp}$ | -0.48260712 | 0.25997095 | -0.24294259 | -0.33588831 |
| $\beta_{2hp}$ | -0.09399809 | 0.43360852 | -0.00285819 | 0.12975114 |
| $\beta_{2kp}$ | 0.08828864 | 0.33102009 | 0.12017906 | 0.11112443 |
| $\beta_3$ | 0.02919402 | 0.04245054 | 0.01179556 | 0.04770020 |

BG, β-glucosidase; NAG, *N*-acetylglucosaminidase; LAP, leucine aminopeptidase; AP, acid phosphatase

---

## Author Response (AR1)

Dear Editor Prof. Yakov Kuzyakov,

Thank you very much for your positive feedbacks and for giving us an opportunity to resubmit the revised manuscript. Based on the reviewers' constructive comments, we have strongly improved this manuscript. We hope the revised manuscript will meet the standards of your journal Biogeosciences. Please see our detailed point-by-point responses below.

Best regards,

Xiaoqi Zhou on behalf of all authors

East China Normal University, Shanghai 200241, China

Editor's comments to the Author:

Dear Authors,
the reviewers mentioned important shortcomings you need to improve before acceptance.
R: Done as suggested.

Additionally, please improve the following:
- Add some quantitative results/Conclusions to your Abstract.
R: Done as suggested.
   The results showed that incubation temperature and tree species significantly influenced all soil EEA and *Eucalyptus* had 1.01-2.86 times higher soil EEA than coniferous tree species. Modeling showed that *Eucalyptus* had larger soil C losses but had 0.99-2.38 times longer soil C residence time than the coniferous tree species over time.
   On the other hand, the modeling results help explain why exotic slash pine can grow faster, as it has 1.22-1.38 times longer residual soil N residence time for LAP, which mediate soil N cycling in the long term, than native coniferous tree species like hoop pine and kauri pine (*Agathis robusta* C. Moore).
   Page 1 Lines 23-24 and Line 28

- Because the modelling is the most important part of your paper - please explain the models with more details

R: Done as suggested.

We assumed that the differences in soil properties and litter C/N contents under different tree species are the results of effects of tree species, and therefore we established a new soil–enzyme–C/N model to consider the effects of both tree species and incubation temperature without considering other soil properties and litter C inputs derived from tree species. In other words, we considered changes in soil properties and C inputs to be a 'black box' as part of the overall effects of tree species, all of which influenced soil EEA.

We first transformed the enzyme activity data using a natural logarithm. As the enzyme activity data for each plot were not independent along a gradient of temperatures, we needed to consider the interaction of tree species and incubation temperature on soil EEA.

We found that the interactions between incubation temperature and tree species were not significant on soil EEA (Table S1). Therefore based on the **Model 1**, we further established a simpler model (**Model 2**) without considering their interactions.

A conventional soil enzyme–C model (**Model 3**) (Schimel and Weintraub, 2003) has been widely used to predict how soil organic C contents change with soil EEA over time.

In this study, for quantitative analysis of the changes in total C (TC) contents over time under tree species, we combined **Model 2** with the addition of TC and **Model 3** together to establish a dynamic tree species–enzyme–C model (**Model 4**) as shown below:

To get a better understanding these 4 models, we made a simple table to compare the advantages and disadvantages of each model (Table 2).

Page 4 Lines 32-33, Page 5 Lines 1-6, 22-23, 32-33, Page 6 Lines 1-3, 7-8

- Add assumptions to the models

R:Done as suggested.

We assumed that the differences in soil properties and litter C/N contents under different tree species are the results of effects of tree species, and therefore we established a new soil–enzyme–C/N model to consider the effects of both tree species and incubation temperature without considering other soil properties and litter C inputs derived from tree species. In other words, we considered changes in soil properties and C inputs to be a 'black box' as part of the overall effects of tree species, all of which influenced soil EEA.

Page 4 Lines 32-33 and Page 5 Lines 1-3

- to get a better overview about the 4 models - make a simple Table describing the main differences, advantages/disadvantages of each model

R:Done as suggested.

**Table 2.** Description and comparison of the 4 models used in this study.

| | Model 1 | Model 2 | Model 3 | Model 4 |
|---|---|---|---|---|
| Advantage | Tree species–enzyme model with considering the effects of tree species, incubation temperature and their interactions on soil EEA | Tree species–enzyme model with only considering tree species and incubation temperature on soil EEA, as their interactions were not significant in this study (see Table S1) | Conventional enzyme–C model | Tree species–enzyme–C model by combining **Model 2** and **Model 3** |
| Disadvantage | Without considering the interactions of soil EEA | Without considering the interactions of soil EEA | Without considering the effects of tree species and without considering the interactions of soil EEA | Without considering the interactions of soil EEA |

Page 14 Line 7

- Check for papers of Razavi BS, Blagodatskaya E - they made important steps for understanding of temperature effects on soil enzymes; Especially to the jumps you have between 15 and 20 °C (e.g. in Fig 1)

R: Done as suggested.

Interestingly, we noticed that for a certain tree species, the gaps between residual soil C contents with BG at $23^{o}C$, $25^{o}C$ and $27^{o}C$ increased with time, which may be explained by the canceling effects (absence or strong reduction of response of the enzyme to temperature) of soil EEA (Razavi et al., 2015). Previous findings showed that this phenomena was most pronounced at low substrate concentrations (Razavi et al., 2015), which was consistent with our results in Fig. 3.

   Page 7 Lines 6-10

- Move Tables 2 and 3 to Supplementary

R:Done as suggested.

- What are the units on X axis on Fig 2-6? --> these figs are not clear at all.

R: We assume that the current differences in soil properties and litter C/N contents to be a 'black box' and are mainly derived from the effects of tree species. We therefore established a soil–enzyme–C/N model with both tree species and incubation temperature without considering the influence of litter C inputs and soil pH etc on soil

EEA. Our model in this study is an ideal model without a unit for decomposition time. However, we have provided the half-residence times of residual soil C and N contents under different tree species in Table S6 and Table S7.

Additionally, we have re-drawn these Figs to make them clearer. Please check them below.

[Figure]

**Figure 2.** Residual soil C contents under different tree species across time for β-glucosidase (BG), *N*-acetylglucosaminidase (NAG), leucine aminopeptidase (LAP) and acid phosphatase (AP) at 25°C. The total soil C decomposition over time was calculated via Equation 5 and the residual soil C contents over time was compared for different enzyme activities among the tree species.